# Data-Driven Models for Gas Turbine Online Diagnosis

Iván González Castillo [1], Igor Loboda [2,*] and Juan Luis Pérez Ruiz [3]

1   Secretaría de Marina Armada de México, Centro de Mantenimiento Aeronaval del Golfo, Carretera Xalapa—Veracruz, km 6.5, Col. Las Bajadas, Veracruz 91698, Mexico; chrivangc@gmail.com

2   Instituto Politécnico Nacional, Escuela Superior de Ingeniería Mecánica y Eléctrica, Av. Santa Ana, 1000, Mexico City 04430, Mexico

3   Unidad de Alta Tecnología-Facultad de Ingeniería, Universidad Nacional Autónoma de México, Fray Antonio de Monroy e Híjar 260, Juriquilla, Queretaro City 76230, Mexico; perezruiz305@gmail.com

\*   Correspondence: igloboda@gmail.com

**Abstract:** The lack of gas turbine field data, especially faulty engine data, and the complexity of fault embedding into gas turbines on test benches cause difficulties in representing healthy and faulty engines in diagnostic algorithms. Instead, different gas turbine models are often used. The available models fall into two main categories: physics-based and data-driven. Given the models' importance and necessity, a variety of simulation tools were developed with different levels of complexity, fidelity, accuracy, and computer performance requirements. Physics-based models constitute a diagnostic approach known as Gas Path Analysis (GPA). To compute fault parameters within GPA, this paper proposes to employ a nonlinear data-driven model and the theory of inverse problems. This will drastically simplify gas turbine diagnosis. To choose the best approximation technique of such a novel model, the paper employs polynomials and neural networks. The necessary data were generated in the GasTurb software for turboshaft and turbofan engines. These input data for creating a nonlinear data-driven model of fault parameters cover a total range of operating conditions and of possible performance losses of engine components. Multiple configurations of a multilayer perceptron network and polynomials are evaluated to find the best data-driven model configurations. The best perceptron-based and polynomial models are then compared. The accuracy achieved by the most adequate model variation confirms the viability of simple and accurate models for estimating gas turbine health conditions.

**Keywords:** inverse models; data-driven models; multilayer perceptron; polynomials; GasTurb

## 1. Introduction

Reliable gas turbine health information is essential for successful the implementation of condition-based maintenance [1] Gas-path diagnostic techniques provide such information by analyzing engine performance and early identifying potential faults before they develop into serious accidents [2]. Many maintenance actions are based on these diagnostic judgments resulting in maximal asset profitability, reliability, and availability and minimal life cycle costs [3]. To get information about engine operation and possible faults, the diagnostic algorithms employ recorded data and different models. According to the utilization of the data and the models, the gas-path diagnostics can be split into two main approaches.

The first approach, called a Gas Path Analysis (GPA), relies on a nonlinear physics-based engine model. Such a model, also known as a thermodynamic model, involves thermodynamic and other physical relationships to compute gas path variables for given engine operating conditions and health parameters. Engine components (compressor, turbine, burner, etc.) are presented in the model by experimental component performance maps initially corresponding to a new engine. As with real faults that change component performances, the health parameters can slightly shift component performance maps to

simulate different degradation mechanisms. Typically, the flow capacity and efficiency performances of each component are corrected [4]. GPA can also employ a linear model, however, to compute the necessary fault influence coefficient matrix, the thermodynamic model is still used.

Known thermodynamic model programs include but are not limited to Gas turbine Simulation Program (GSP), Numerical Propulsion System Simulation (NPSS), Propulsion Object-Oriented Simulation Software (PROOSIS), and GasTurb [3,5]. The latter software was developed by Kurzke [6] and offers nonlinear simulation for many configurations of different gas turbine engines [3]. A nonlinear thermodynamic model enables the creation of different simplified models. The most known of them is a linear model that relates small changes of health parameters to the corresponding changes of monitoring variables through a so-called influence coefficient matrix.

The described nonlinear and linear models present a direct mathematical problem, in which operating conditions and health parameters are independent quantities and monitored variables present dependent quantities. The considered diagnostic approach aims to solve an inverse problem [7] that allows estimating unmeasured parameters through the measured variables and knowledge of the physical processes of the system, making inferences from data [8]. A difficulty of inverse problems on multidimensional space is the instability of their solutions, in other words, small errors in measurements can cause large errors in the estimated parameters. There exist many methods to solve inverse problems, among them, regularization, Truncated Singular Value Decomposition (TSVD), interactive methods, discretization, the maximum entropy method, Algebraic Reconstruction Technique (ART), and the Backus Gilbert method [7].

In the case of GPA, an inverse problem consists in evaluating the conditions of engine components through estimating health parameters using measured operating conditions and monitored variables [9]. In this way, a general fleet-average model is adapted to a particular real engine, and this adaption problem belongs to the area of system identification. When diagnostic algorithms employ the linear model, diagnostic matrixes [9] are computed, for example, by the least squares method [9,10]. Study [11] reveals that linearization significantly increases the errors of estimated parameters. If engine diagnosis relies on a nonlinear thermodynamic model, different iterative procedures of the model identification are applied to estimate health parameters. For example, the authors of paper [12] propose and employ the approach called adaptive modeling, paper [13] performs the nonlinear model identification using a genetic algorithm, paper [14] proposes multicriteria identification, and paper [15] presents a strategy for automatic adaption of an identification scheme in the case of sensor malfunctions.

The second main diagnostic approach called data-driven relies on available real data and pattern recognition techniques. This approach consists of three stages: data acquisition, preliminary data processing or feature extraction, and a diagnostic process itself. Data can be acquired in three ways: field data recording, experimental test bed data acquisition, and data generation by an engine model. The field recordings suffer the data with engine fault manifestations, and physical fault embedding in real engines in test beds is too expensive [16]. Therefore, the simulation of faults-affected engine data is an effective and widely used option.

The stage of feature extraction determines the features that are sensitive to engine faults. Typically, deviations of measured values of monitored variables from baseline values corresponding to a healthy engine are computed. Given that the engine variables depend on an operating point, a baseline model is determined through the approximation of available healthy engine data at different operating points. Such a model can be called data-driven or "black box". It does not need detailed knowledge of engine functioning and can be easily created by gas turbine operators using, for example, polynomial regression [1] or multilayer perceptron [17] as approximation functions.

Deviations of monitored variables form a feature vector, which is a pattern to be recognized. From such patterns a classification is created, in which each class is presented

by numerous patterns corresponding to a specific fault. The classification data allow learning a recognition technique, mostly one of artificial neural networks.

The present paper proposes and proves a new simulation methodology for the GPA that originally presents a physics-based approach. The nonlinear thermodynamic model is accurate but complex and critical to computer resources. On the other hand, the linear model is fast but inaccurate because of linearization errors, which significantly intensify in the inverse procedure of estimating health parameters. It is proposed to consolidate the advantages of these models by creating a new nonlinear data-driven model that computes health parameters using measured operating conditions and monitored variables as inputs. Because of its nonlinear nature, such an inverse diagnostic model can conserve the accuracy of the underlying thermodynamic model. On the other hand, it will be fast due to simple data-driven functions involved and no need in inverse procedures to diagnose an engine. Furthermore, the new model will be by far more stable and reliable because during iterative computing of the thermodynamic model an engine operating point can go beyond components performance maps resulting in the loss of convergence and even abnormal program termination. The proposed inverse nonlinear data-driven model and the corresponding mode to diagnose gas turbines present a hybrid approach because the original GPA is now based on a data-driven model. Such an approach to engine modeling and diagnosis is not mentioned in extensive reviews on gas turbine diagnostics [2,3], and to our knowledge, it is novel.

To realize and prove the proposed approach, this paper employs two aircraft engines of different types, namely, turbo shaft and turbo fan. For each engine, polynomial-based and multilayer perceptron-based variations of the inverse model were created. The necessary data to build and test the models were generated by the software GasTurb that offers a nonlinear thermodynamic simulation of main gas turbine types. The use of this well-known and commercially available software in the present study allows every researcher to repeat the investigation and verify its results. For such a verification the paper provides all necessary details of the calculations performed.

## 2. Gas Turbine Simulation Techniques

### 2.1. Thermodynamic Model

As it follows from the previous section, the data required to create the inverse non-linear models are generated in GasTurb by gas turbine thermodynamic models of the corresponding test-case engines. Each gas turbine component is presented in a thermodynamic model by a component performance map. Using known thermodynamic relations and mechanical laws, this model relates gas path variables to component performances and engine steady-state operating points. The process of computing gas path variables is organized as the solution of a system of nonlinear algebraic equations reflecting a balance of mass, heat, and energy in those components at steady states. The system is usually solved by the Newton–Raphson method, aka Newton's method. The computation is complex and includes some iterative cycles. The model software is considerably complex and comprises dozens of subprograms. As a result of the computation, the model determines gas path quantities including an $[m \times 1]$-vector $\vec{Y}$ of monitored variables as a function of an $[n \times 1]$-vector $\vec{U}$ of operating conditions (power set and ambient variables) and an $[r \times 1]$-vector $\vec{\Theta}$ of special health parameters. Thus, the thermodynamic model can be presented by the following mathematical expression [18]:

$$\vec{Y} = f(\vec{\Theta}, \vec{U}) \tag{1}$$

The health parameters can be presented by an expression $\vec{\Theta} = \vec{\Theta}_0 + \delta\vec{\Theta}$. A vector $\vec{\Theta}_0$ corresponds to a healthy engine and nominal component performances, while a vector $\delta\vec{\Theta}$ of fault parameters allows shifting performance maps to simulate different scenarios of engine deterioration with varying severity.

### 2.2. Multilayer Perceptron

The multilayer perceptron (MLP) is known as an artificial neural network capable to solve both classification and approximation problems. As an approximation technique, it is employed in gas turbine diagnostics for building a gas turbine baseline model [17] that computes monitored variables as a function of operating conditions. Figure 1 illustrates the structure and operation of the MLP [19,20] proposed to develop the diagnostic models for the present study. MLP includes three types of layers. The input layer is represented by operating conditions and monitored variables and is followed by two hidden layers and an output layer of the health parameters.

The perceptron presents a feed-forward network in which all signals go from the input to the output. In this network, all nodes of adjacent layers are connected. Each connection has a weight coefficient. The coefficient of one layer constitute the corresponding matrix $W_1$, $W_2$, or $W_3$. All the signals of one layer multiplied by its weight coefficients and summed to form an input to each node of the subsequent layer. A transfer function $f_1$, $f_2$, or $f_3$ transforms this input into a node output. The outputs of all the layer nodes constitute a vector $\vec{a}_1$, $\vec{a}_2$, or $\vec{a}_3$. Then these operations are repeated for the next layer, and so on. Finally, output signals are computed. The described procedure can be written by the following expressions $\vec{a}_1 = f_1(W_1\vec{p})$, $\vec{a}_2 = f_2(W_2\vec{a}_1)$, and $\delta\vec{\Theta} = \vec{a}_3 = f_3(W_3\vec{a}_2)$, where $\vec{p}$ is a network input vector constituted by operating conditions $U$ and monitored variables $Y$.

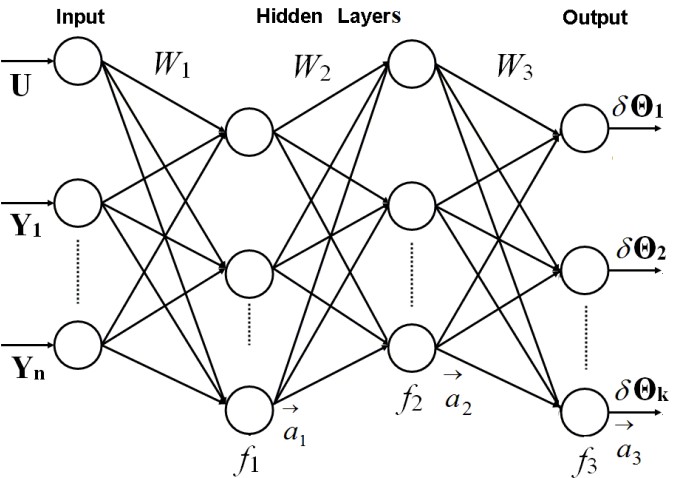

**Figure 1.** Two hidden layers MLP proposed to develop the GasTurb diagnostic models.

To find unknown weight coefficients, the perceptron undergoes multiple iterations (epochs) of a back-propagation learning algorithm that minimizes the total error of the network output. In every epoch, the coefficients are corrected in the direction that reduces the output error. The epochs are repeated unless the algorithm finds the error minimum. Before estimating waiting coefficients, network hyperparameters should be determined, namely, the number of hidden layers, numbers of neurons on each hidden layer, type of activation functions, and number of epochs. This is an optimization process [21] usually conducted through manual trial-and-error methods [22].

### 2.3. Polynomials

The research conducted by Loboda et al. [23] successfully applies second-order polynomials to create a gas turbine baseline model. Hence, this type of polynomials was chosen as a candidate technique in the present study.

As described in Section 1, the proposed inverse diagnostic model has structure $\vec{\Theta}(\vec{Y},\vec{U})$. The model was created on the data corresponding to standard ambient conditions. Therefore, within the vector $\vec{U}$ the model uses only one element, a power set variable. To use the

model for other ambient conditions, the input variables $\vec{Y}$ and $U$ should be corrected to the standard conditions using known correction relations [24]. Given the above explanation, the second-order polynomial model for one fault parameter $\delta\Theta$ takes a form:

$$
\begin{aligned}
\delta\Theta\left(\vec{Y},U\right) = {} & a_1 + a_2 Y_1 + a_3 Y_2 + a_4 Y_3 + a_5 Y_4 + a_6 Y_5 + a_7 Y_6 + a_8 U + a_9 Y_1 Y_2 + a_{10} Y_1 Y_3 \\
& + a_{11} Y_1 Y_4 + a_{12} Y_1 Y_5 \ldots + a_{k-1} Y_6{}^2 + a_k U^2
\end{aligned}
\tag{2}
$$

This equation can be rewritten in a vector form as $\delta\Theta = \vec{V}^T \vec{A}$. Where $\vec{A}$ is a $k \times 1$ vector of the coefficients, and $\vec{V}$ presents a $k \times 1$ vector of products of $Y$ and $U$ multiplied by the corresponding coefficients in Equation (2).

The polynomials for all fault parameters $\vec{\delta\Theta}$ are presented in a generalized form by the following expression:

$$
\vec{\delta\Theta}^T = \vec{V}^T A
\tag{3}
$$

where a $k \times r$ matrix A includes unknown coefficients $a_i$ for all $r$ health parameters. Given that a single operating point is not sufficient to compute all the coefficients, the data used to estimate the coefficients incorporate $t$ operating points with different operating and health conditions. Let the totality of the health parameters values be a $t \times r$ matrix $\delta\Theta$ and the set of all the values of $Y$ and $U$ products be a $t \times k$ matrix V. A linear system of equations for parameters $\delta\Theta$ will take a form $\delta\Theta =$ VA then.

For the sake of high accuracy of estimates $\delta\Theta$, a great volume of the input data is engaged, and a widespread solution

$$
\hat{A} = \left(V^T V\right)^{-1} V^T \delta\Theta
\tag{4}
$$

is drawn by the least-squares method.

The input data were generated by the software GasTurb for turbo shaft and turbo fan engines. The turbo shaft is set as a primary test case engine and has great volume of generated data, on which, many influencing factors are studied. The turbo fan is a secondary test case engine with smaller volume of data to verify a general viability of diagnostic models proposed.

### 3. Turbo Shaft Diagnostic Models

*3.1. Input Data*

For the turbo shaft, GasTurb simulates the following variables that are usually measured in real engines and can be used for engine monitoring: temperatures and pressures at the outlet of each compressor and turbine and fuel flow. Among the operating conditions, a compressor spool speed, ambient temperature, and ambient pressure are simulated. The present study uses the spool speed as a model input variable while the ambient temperature and pressure have constant standard values. The available and chosen fault parameters are those that correct capacity and efficiency performances of the compressor and turbines. Table 1 specifies all these quantities.

The multidimensional space of the chosen fault parameters causes the need for a large amount of simulated data to fully cover this space. Therefore, each of the six fault parameters ΔCC, ΔCE, ΔHPTC, ΔHPTE, ΔPTC, and ΔPTE had the same 6 levels of values: 0%, 1%, 2%, 3%, 4%, and 5%. These fixed levels were introduced to better see the influence of degradation severity on simulation errors. Five operating regimes were set by different spool speed values. For each regime, the simulated data include the necessary $Y$ and $U$ variables at 1458 operating points with different combinations of the fault parameters and their levels. In total for all operating regimes, the data embrace 7290 operating points. They are divided into a learning set embracing 85% of data from each regime and a validation set with the rest of the data.

For additional model verification, a testing set was also generated. At each operating regime it has 150 points with a random distribution of the fault parameters in the interval (0%, −5%), resulting in a total of 750 points for the 5 regimes. This uniform random parameter distribution allows fully covering a simulation area and therefore is favorable to accurate data approximation. The chosen parameter interval corresponds to the recommendations of the extensive study carried out by Fentaye et al. [2] on different degradation mechanisms and engine performance losses that they cause.

**Table 1.** Turbo shaft simulated quantities.

| # | Name | Abbreviation |
|---|------|-------------|
| | OPERATING CONDITION ($U$) | |
| 1 | Compressor Spool Speed | ZXN |
| | FAULT PARAMETERS ($\Theta$) | |
| 1 | Compressor Capacity Delta [%] | $\Delta$CC |
| 2 | Compressor Efficiency Delta [%] | $\Delta$CE |
| 3 | HPT Capacity Delta [%] | $\Delta$HPTC |
| 4 | HPT Efficiency Delta [%] | $\Delta$HPTE |
| 5 | PT Capacity Delta [%] | $\Delta$PTC |
| 6 | PT Efficiency Delta [%] | $\Delta$PTE |
| | MONITORED VARIABLES ($Y$) | |
| 1 | Shaft Power Delivered | SPD |
| 2 | Fuel Flow | FF |
| 3 | Compressor Exit Pressure | P3 |
| 4 | HP Turbine Exit Pressure | P44 |
| 5 | PT Turbine Exit Pressure | P5 |
| 6 | Compressor Exit Temperature | T3 |
| 7 | HP Turbine Exit Temperature | T44 |
| 8 | PT Turbine Exit Temperature | T5 |

### 3.2. Perceptron Configuration

Perceptron with one hidden layer architecture is sufficient for any classification problem, and several studies confirm it [25]. However, for a continuous function approximation problem, one hidden layer did not prove to be better than a second-degree polynomial [23]. Based on this reasoning and studies on noise filtering [26,27] which recommend a two hidden layers perceptron, configurations with one and two layers were chosen for the proposed diagnostic models. In these models, the monitored variables $\vec{Y}$ and the operating condition $U$ are inputs, and the fault parameters $\vec{\delta\Theta}$ are outputs. Kolmogorov's theorem is a notable reference for an initial perceptron configuration, proposing that the number of neurons in the hidden layer of an ANN should not be greater than $2n + 1$ [28], where $n$ denotes the number of inputs. The perceptron is learned using the variations of a back-propagation algorithm [29] available in Matlab.

### 3.3. One Regime Diagnostic Model

To gain better understanding of the influence of the arguments on model accuracy, a simplified model $\vec{\delta\Theta}(\vec{Y})$ for diagnosing on one fixed regimen is first created. The MLP network employed for this model has the monitored variables $\vec{Y}$ as inputs and the fault parameters $\vec{\delta\Theta}$ as outputs. Different configurations of one and two hidden layers were tested, using all the 18 training methods available in Matlab. For each method, the learning was performed with different numbers of training epochs to choose the optimal number for the method. The comparison criterion for the analyzed methods was the root mean square error (RMSE) between the true values used in GasTurb and the values simulated by the network. Table 2 shows the results of the best final configurations of the network with one hidden layer (ANN 1) and the network with two hidden layers (ANN 2). The operating regime is set here by a relative spool speed of 1.0 that corresponds to the maximum power

of 100%. One can see that, according to the average accuracy shown in the last row, ANN 2 trained by the trainbr Matlab function (Bayesian Regulation) is the best configuration.

**Table 2.** RMSE for MLP configurations of one regime diagnostic model (turbo shaft, relative spool speed 1.0).

| Fault Parameters | ANN 1 Training Method: Trainrp Epochs: 8000 | | ANN 2 Training Method: Trainrp Epochs: 15,000 | | ANN 1 Training Method: Trainbr Epochs: 500 | | ANN 2 Training Method: Trainbr Epochs: 500 | |
|---|---|---|---|---|---|---|---|---|
| | Learning | Validation | Learning | Validation | Learning | Validation | Learning | Validation |
| ΔCC | 0.1910 | 0.1992 | 0.0380 | 0.0400 | 0.0109 | 0.0116 | 0.0073 | 0.0081 |
| ΔCE | 0.1037 | 0.1092 | 0.0163 | 0.0184 | 0.0076 | 0.0082 | 0.0073 | 0.0079 |
| ΔHPTC | 0.2043 | 0.2062 | 0.1797 | 0.1811 | 0.0205 | 0.0228 | 0.0089 | 0.0091 |
| ΔHPTE | 0.1974 | 0.2019 | 0.0387 | 0.0415 | 0.0151 | 0.0159 | 0.0085 | 0.0087 |
| ΔPTC | 0.2051 | 0.2125 | 0.1783 | 0.1843 | 0.0246 | 0.0274 | 0.0076 | 0.0078 |
| ΔPTE | 0.1971 | 0.2016 | 0.0592 | 0.0614 | 0.0286 | 0.0304 | 0.0078 | 0.0088 |
| Average | 0.1831 | 0.1884 | 0.0850 | 0.0878 | 0.0179 | 0.0194 | 0.0079 | 0.0084 |

After the demonstration of the high accuracy of the two hidden layers MLP, tests were performed to determine the number of neurons in the second hidden layer. Different numbers were tested keeping constant the regime (100%), eight neurons in the first hidden layer, the method (Bayesian Regulation), and the training epochs (500). Table 3 shows the results, which allow pointing out that increasing the neurons number generally enhances the accuracy, and the best network configuration has 28 hidden neurons.

**Table 3.** Selection of optimal neurons number (turbo shaft, relative spool speed 1.0).

| Fault Parameter | Neurons in the 2nd Hidden Layer | | | | | | | | | | |
|---|---|---|---|---|---|---|---|---|---|---|---|
| | 9 | 10 | 11 | 12 | 13 | 14 | 15 | 16 | 17 | 18 | 19 |
| ΔCC | 0.1809 | 0.0290 | 0.0339 | 0.0301 | 0.0319 | 0.0174 | 0.0198 | 0.0124 | 0.0100 | 0.0166 | 0.0099 |
| ΔCE | 0.0367 | 0.1811 | 0.0252 | 0.0172 | 0.0221 | 0.0157 | 0.0169 | 0.0153 | 0.0162 | 0.0073 | 0.0156 |
| ΔHPTC | 0.1767 | 0.0478 | 0.1754 | 0.1500 | 0.1789 | 0.0190 | 0.0177 | 0.0125 | 0.0197 | 0.0117 | 0.0122 |
| ΔHPTE | 0.0346 | 0.0398 | 0.0391 | 0.0241 | 0.0388 | 0.0165 | 0.0174 | 0.0191 | 0.0129 | 0.0115 | 0.0071 |
| ΔPTC | 0.0445 | 0.1828 | 0.0392 | 0.1519 | 0.0426 | 0.0200 | 0.0186 | 0.0231 | 0.0164 | 0.0187 | 0.0114 |
| ΔPTE | 0.0336 | 0.1836 | 0.1813 | 0.0464 | 0.0288 | 0.0160 | 0.0257 | 0.0149 | 0.0134 | 0.0128 | 0.0203 |
| Average | 0.0845 | 0.1107 | 0.0824 | 0.0700 | 0.0572 | 0.0174 | 0.0193 | 0.0162 | 0.0148 | 0.0131 | 0.0128 |

| Fault Parameters | Neurons in the 2nd Hidden Layer | | | | | | | | | | |
|---|---|---|---|---|---|---|---|---|---|---|---|
| | 20 | 21 | 22 | 23 | 24 | 25 | 26 | 27 | 28 | 29 | 30 |
| ΔCC | 0.0084 | 0.0081 | 0.0090 | 0.0081 | 0.0070 | 0.0082 | 0.0101 | 0.0073 | 0.0055 | 0.0066 | 0.0089 |
| ΔCE | 0.0069 | 0.0057 | 0.0067 | 0.0079 | 0.0050 | 0.0052 | 0.0055 | 0.0066 | 0.0044 | 0.0055 | 0.0063 |
| ΔHPTC | 0.0086 | 0.0083 | 0.0109 | 0.0091 | 0.0082 | 0.0086 | 0.0087 | 0.0084 | 0.0059 | 0.0068 | 0.0090 |
| ΔHPTE | 0.0084 | 0.0088 | 0.0126 | 0.0087 | 0.0075 | 0.0082 | 0.0083 | 0.0106 | 0.0065 | 0.0073 | 0.0064 |
| ΔPTC | 0.0098 | 0.0099 | 0.0112 | 0.0078 | 0.0085 | 0.0100 | 0.0114 | 0.0087 | 0.0068 | 0.0074 | 0.0091 |
| ΔPTE | 0.0098 | 0.0084 | 0.0135 | 0.0088 | 0.0096 | 0.0093 | 0.0107 | 0.0097 | 0.0064 | 0.0086 | 0.0088 |
| Average | 0.0087 | 0.0082 | 0.0107 | 0.0084 | 0.0076 | 0.0082 | 0.0091 | 0.0085 | 0.0059 | 0.0071 | 0.0081 |

Once the number of neurons in the second hidden layer was defined, the MLP was also tested at the other 4 operating regimes given by relative spool speed values 0.9 (90%), 0.8, 0.7, and 0.6. The results for all 5 regimes presented in Table 4 show that an accuracy level remains high for all the regimes.

**Table 4.** RMSE for 5 operating regimes.

| Fault Parameter/Relative Compressor Spool Speed | 1 | 0.9 | 0.8 | 0.7 | 0.6 |
|---|---|---|---|---|---|
| ΔCC | 0.0055 | 0.0095 | 0.0066 | 0.0105 | 0.0075 |
| ΔCE | 0.0044 | 0.0064 | 0.0050 | 0.0083 | 0.0054 |
| ΔHPTC | 0.0059 | 0.0086 | 0.0084 | 0.0084 | 0.0090 |
| ΔHPTE | 0.0065 | 0.0114 | 0.0075 | 0.0077 | 0.0044 |
| ΔPTC | 0.0068 | 0.0114 | 0.0084 | 0.0098 | 0.0113 |
| ΔPTE | 0.0064 | 0.0123 | 0.0125 | 0.0160 | 0.0166 |
| Average | 0.0059 | 0.0099 | 0.0081 | 0.0101 | 0.0090 |

Figure 2 confirms a high degree of approximation accuracy achieved by the MLP network. Figure 2a shows three fault parameters used in GasTurb in comparison with the same parameters simulated by MLP. Both curves practically coincide in the figure for all parameters in all points. Enlarged scale plot of the second parameter in Figure 2b shows that maximal difference between original and MLP values is about 0.005%. It is 200 times smaller than 1% that is usually considered as the minimal value of the changes induced by engine faults. Thus, one can state that the MLP-based model of fault parameters is accurate enough and useful for diagnosing engines.

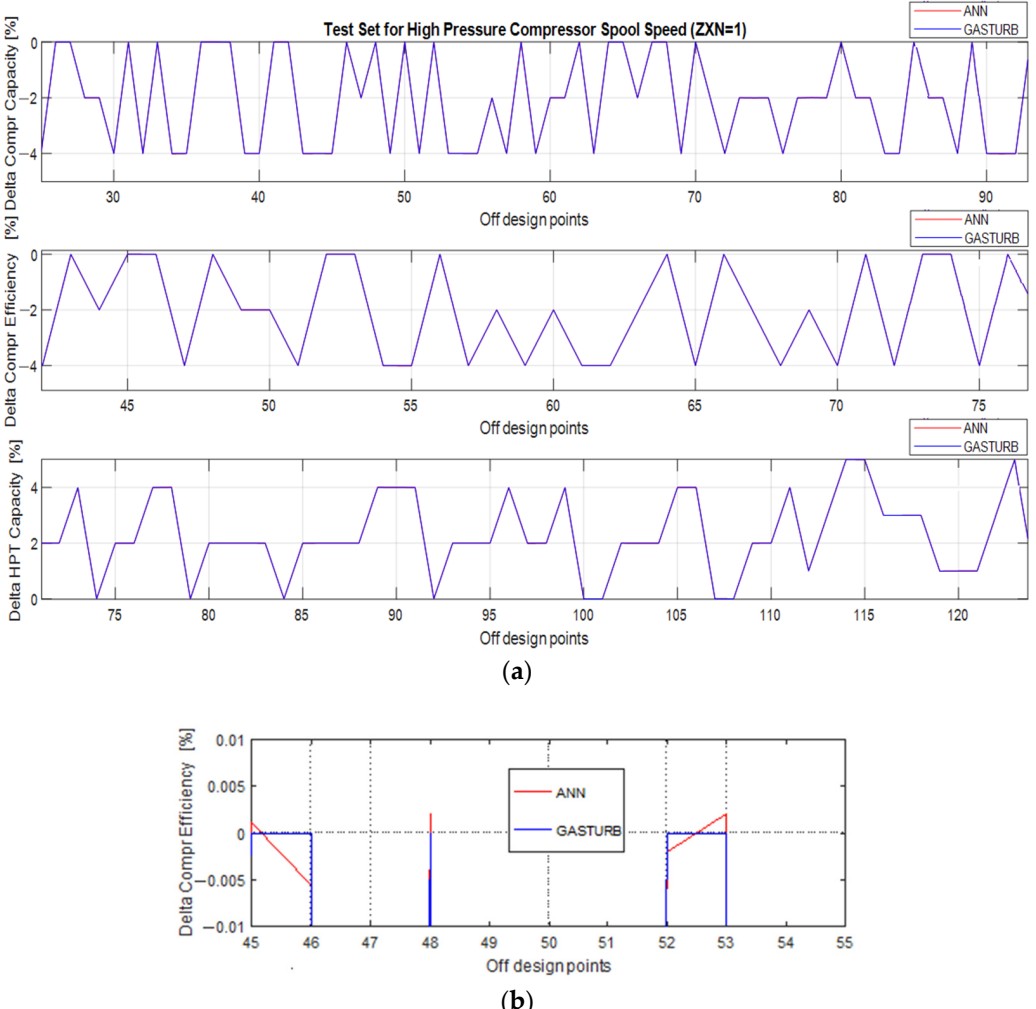

(**a**)

(**b**)

**Figure 2.** Original fault parameters and parameters simulated by ANN-based one regime diagnostic model ((**a**) 3 parameters, normal scale; (**b**) compressor efficiency parameter, enlarged scale).

### 3.4. Extended Diagnostic Model

The model studied in the previous subsection is intended for diagnosing on a fixed regime. To make the model more universal and useful, it is extended by adding a spool speed variable to the model inputs. Such a model will have a structure $\delta\vec{\Theta}(\vec{Y},U)$. In contrast to the previous subsection, where each model was determined with an individual data set generated at one of the 5 regimes, the new model will be learned with all data sets available from different regimes. Since the model change is small, from 9 to 10 inputs with the same outputs, the same MLP configurations were verified. The one- and two-hidden-layers MLP were again tested with a wide variety of neuron numbers, epochs, and training methods. Table 5 shows the test results for two network configurations and two learning functions. As before, two hidden layer networks learned by the Bayesian training method expose the best accuracy.

**Table 5.** RMSE for best MLP configurations of extended diagnostic model.

| Fault Parameter | ANN 1 Trainrp | | ANN 2 Trainrp | | ANN 1 Trainbr | | ANN 2 Trainbr | |
|---|---|---|---|---|---|---|---|---|
| | LEARN | VALID | LEARN | VALID | LEARN | VALID | LEARN | VALID |
| ΔCC | 0.1086 | 0.103 | 0.0995 | 0.101 | 0.0146 | 0.0151 | 0.0499 | 0.0502 |
| ΔCE | 0.2582 | 0.257 | 0.2515 | 0.2447 | 0.1171 | 0.117 | 0.0446 | 0.0465 |
| ΔHPTC | 0.203 | 0.2064 | 0.2171 | 0.2185 | 0.0545 | 0.0602 | 0.0416 | 0.0451 |
| ΔHPTE | 0.2401 | 0.2394 | 0.2469 | 0.2476 | 0.0857 | 0.0856 | 0.0476 | 0.0478 |
| ΔPTC | 0.2143 | 0.2084 | 0.225 | 0.2142 | 0.043 | 0.0457 | 0.0663 | 0.0721 |
| ΔPTE | 0.2783 | 0.2846 | 0.2636 | 0.2713 | 0.1779 | 0.1798 | 0.18 | 0.1822 |
| Average | 0.2171 | 0.2165 | 0.2173 | 0.2162 | 0.0821 | 0.0839 | 0.0717 | 0.0740 |

Having demonstrated the greater precision of the two-hidden-layer MLP, tests were carried out to determine the number of neurons in the second layer, keeping constant the training method and epochs number. Based on previous experience with the one-regime model, 19 to 31 neurons were probed in the second hidden layer. Table 6 shows the test set results, namely RMSE of health parameters for each node number. As can be seen, the configuration with 27 neurons in the second hidden layer is the most accurate.

**Table 6.** Selection of the optimal neurons number (turbo shaft, extended model).

| Fault Parameters | Neurons in the 2nd Hidden Layer (RMSE) | | | | | |
|---|---|---|---|---|---|---|
| | 19 | 21 | 22 | 23 | 24 | 25 |
| ΔCC | 0.1787 | 0.1796 | 0.0549 | 0.0502 | 0.0438 | 0.0595 |
| ΔCE | 0.0512 | 0.0577 | 0.0427 | 0.0465 | 0.0496 | 0.0451 |
| ΔHPTC | 0.1805 | 0.1771 | 0.0548 | 0.0451 | 0.0396 | 0.0529 |
| ΔHPTE | 0.0453 | 0.0521 | 0.0471 | 0.0478 | 0.0454 | 0.0538 |
| ΔPTC | 0.1822 | 0.0720 | 0.1813 | 0.0721 | 0.0704 | 0.0715 |
| ΔPTE | 0.1799 | 0.1825 | 0.1824 | 0.1822 | 0.1835 | 0.0726 |
| Average | 0.1363 | 0.1202 | 0.0939 | 0.0740 | 0.0721 | 0.0592 |
| Fault Parameters | Neurons in the 2nd Hidden Layer (RMSE) | | | | | |
| | 26 | 27 | 28 | 29 | 30 | 31 |
| ΔCC | 0.0449 | 0.0523 | 0.0464 | 0.0671 | 0.0431 | 0.0455 |
| ΔCE | 0.0373 | 0.0481 | 0.0374 | 0.0445 | 0.0364 | 0.0369 |
| ΔHPTC | 0.0392 | 0.0488 | 0.0406 | 0.0478 | 0.0382 | 0.0399 |
| ΔHPTE | 0.0399 | 0.0491 | 0.0473 | 0.0405 | 0.0424 | 0.0424 |
| ΔPTC | 0.0494 | 0.0617 | 0.0863 | 0.0712 | 0.0522 | 0.0591 |
| ΔPTE | 0.1802 | 0.0601 | 0.1827 | 0.0689 | 0.1808 | 0.1824 |
| Average | 0.0652 | 0.0534 | 0.0734 | 0.0567 | 0.0655 | 0.0677 |

In comparison with the one-regime models that have very low approximation errors, the extended model errors are significantly greater. According to Tables 4 and 6, the errors have increased 5–10 times. Even with this loss of accuracy, the network proposed appears capable of diagnosing. Against the background of fault parameter values that vary from 1% to 5%, the average error will be only 0.0534% i.e., 20 times lower than the smallest parameter value.

### 3.5. Comparison between Approximation Functions for the Extended Model

Loboda et al. [23] compared the approximation capabilities of the second-degree polynomials and a typical single hidden layer MLP in the application to a baseline model. The MLP showed a better data approximation in the training sets, however, in the test sets it lost accuracy, although the authors expected the superiority of the network. They pointed out very similar performances of both tools with slight dominance of the polynomials and abandoned the idea of the superiority of MLP.

The above-cited work inspired us to develop a polynomial-based extended model $\vec{\Theta}(\vec{Y},\vec{U})$ and compare it with the network-based model to choose the best one. Both turbo shaft models were trained on the learning set and applied to the validation and testing sets described in the subsection "Input data".

Figure 3 presents the fault parameters ΔHPTE, ΔPTC, and ΔPTE employed in GasTurb and simulated by second-degree polynomials. All the data plotted in the figure are from the validation set. Figure 4 shows the same parameters, but the simulation was performed by the two-hidden-layer MLP. Comparing these figures separately for each fault parameter, the MLP simulation is much closer to the true values than the polynomial simulation for all the parameters presented.

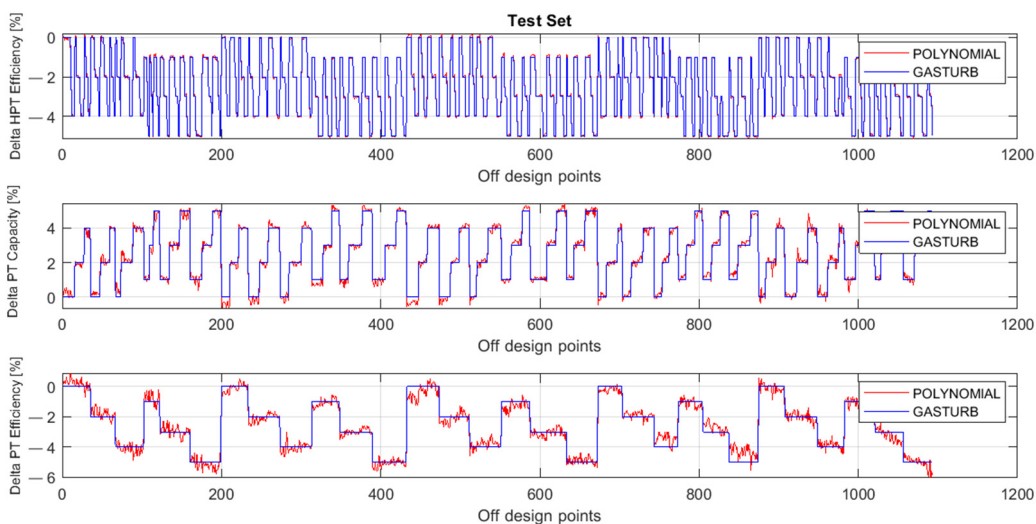

**Figure 3.** Fault parameters ΔHPTE, ΔPTC, and ΔPTE, true and simulated by polynomials (validation set).

Figures 5 and 6 illustrate the behavior of the same parameters as Figures 3 and 4, but the parameters are now computed for the testing set. The comparison of these figures confirms the previous conclusion that the MLP-based model has higher accuracy.

In addition to the qualitative analysis of Figures 3–6, Table 7 helps us to make a quantitative comparison of the polynomials- and the MLP-based models. The table includes mean fault parameter errors of both techniques on the data of the learning, validation, and testing sets. Each data set shows that the MLP-based model is by far more accurate for each parameter and in general.

To validate the results obtained for the turbo shaft, the turbo fan was chosen as the second motor due to its wide use as a power plant for passenger and cargo aircrafts.

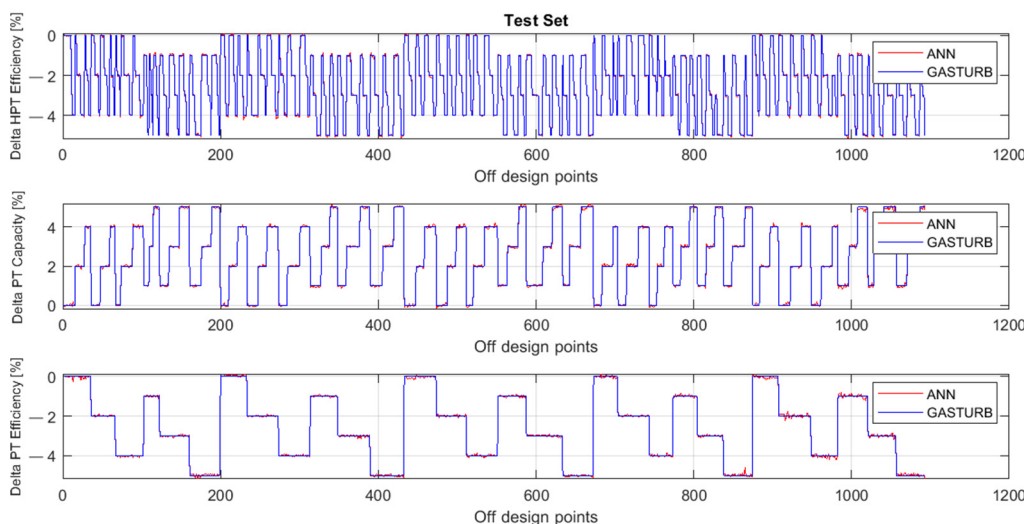

**Figure 4.** Fault parameters ΔHPTE, ΔPTC, and ΔPTE, true and simulated by MLP (Validation set).

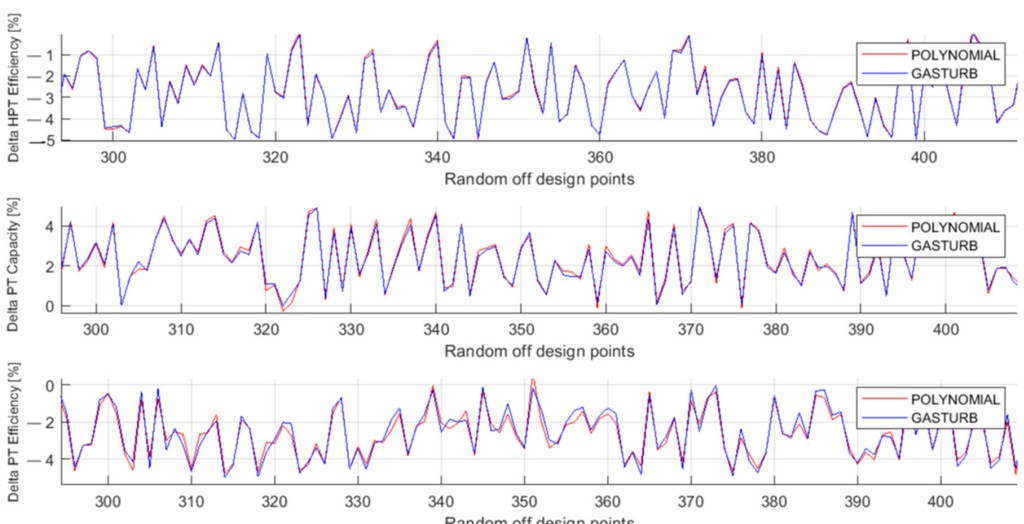

**Figure 5.** Fault parameters ΔHPTE, ΔPTC, and ΔPTE, true and simulated by polynomials (Testing set).

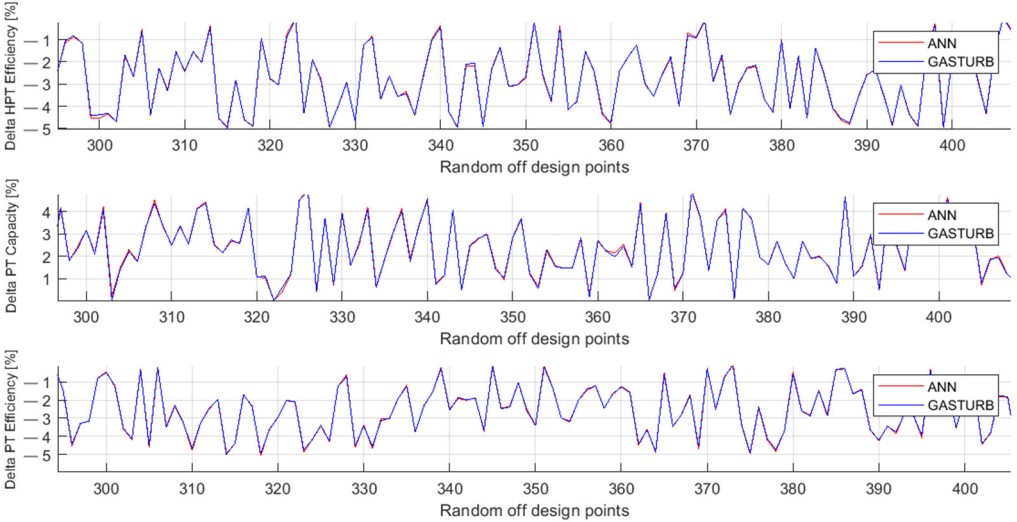

**Figure 6.** Fault parameters ΔHPTE, ΔPTC, and ΔPTE, true and simulated by MLP (Testing set).

**Table 7.** Fault parameter RMSE of polynomials and MLP.

| Fault Parameter | Two Hidden Layers MLP | | | Polynomials | | |
|:---:|:---:|:---:|:---:|:---:|:---:|:---:|
| | **LEARN** | **VALID** | **TEST** | **LEARN** | **VALID** | **TEST** |
| ΔCC | 0.0518 | 0.0523 | 0.0689 | 0.1888 | 0.1849 | 0.1448 |
| ΔCE | 0.0462 | 0.0481 | 0.0570 | 0.0940 | 0.0901 | 0.0693 |
| ΔHPTC | 0.0476 | 0.0488 | 0.0531 | 0.0735 | 0.0715 | 0.0546 |
| ΔHPTE | 0.0505 | 0.0491 | 0.0582 | 0.0664 | 0.0662 | 0.0558 |
| ΔPTC | 0.0593 | 0.0617 | 0.0616 | 0.2287 | 0.2279 | 0.1633 |
| ΔPTE | 0.0593 | 0.0601 | 0.0583 | 0.3946 | 0.3876 | 0.3226 |
| AVERAGE | 0.0525 | 0.0534 | 0.0595 | 0.1743 | 0.1714 | 0.1350 |

## 4. Diagnostic Models of the Turbo Fan

### 4.1. Input Data

As with the turbo shaft, to develop turbo fan diagnostic models, data with different fault and operating conditions were generated by GasTurb. The following variables were simulated and recorded. An engine operating regime is set in GasTurb by an HPC spool speed variable. Among simulated gas path quantities, seven variables usually used for monitoring were chosen. As to fault parameters, their number was increased from six to eight because the turbo fan has a greater complexity and more components that need estimating their health condition. Table 8 specifies all these quantities.

Given that the turbo fan presents a secondary test case aimed to confirm the general viability of the proposed diagnostic model, and that the work with GasTurb is mostly manual, the total volume of generated data is smaller. For each of the 5 regimes, the data include 700 operating points with different fault parameters randomly distributed within the range of 0% to ±5%. In this way, the total data include 3500 operating points. A learning set, which is used to determine turbo fan diagnostic models, embraces 85% of data from each regime. The rest of the data form a validation set intended for the model verification. All the indicators of models' accuracy illustrated in the below subsections were obtained on the validation data.

**Table 8.** Simulated quantities of turbo fan.

| # | Name | Abbreviation | Name | Abbreviation |
|:---:|:---:|:---:|:---:|:---:|
| | **FAULT PARAMETERS (Θ)** | | **MONITORED VARIABLES (*Y*)** | |
| 1 | LPC Capacity Delta [%] | ΔLPCC | Net Thrust | NT |
| 2 | LPC Efficiency Delta [%] | ΔLPCE | Fuel Flow | FF |
| 3 | HPC Capacity Delta [%] | ΔHPCC | HPC Exit Pressure | P3 |
| 4 | HPC Efficiency Delta [%] | ΔHPCE | HPT Exit Pressure | P44 |
| 5 | HPT Capacity Delta [%] | ΔHPTC | LPT Exit Pressure | P5 |
| 6 | HPT Efficiency Delta [%] | ΔHPTE | HPC Exit Temperature | T3 |
| 7 | LPT Capacity Delta [%] | ΔLPTC | LPT Exit Temperature | T5 |
| 8 | LPT Efficiency Delta [%] | ΔLPTE | OPERATING CONDITION (*U*) | |
| | | | HPC Spool Speed | ZXNH |

### 4.2. Extended MLP-Based Turbo Fan Diagnostic Model

Since one regime models of the turbo shaft were intermediate and not intended for a diagnostic application, these models were not created for the turbo fan. The extended models were only formed, and the methodology used to build and verify the turbo shaft models was conserved. As the first step, all available MLP training methods were probed, resulting in choosing the method of Bayesian regularization as the most accurate. Then, different network configurations with a varying number of neurons were tested. The RMSEs for these configurations are placed in Table 9. Comparing it with Table 6, one can observe a general error increase. It is more significant for the first two health parameters. This fact will be further analyzed. As to the influence of a neuron number, it is not strong

now: the average RSME changes from 0.4024 to 0.4122, and the lowest error corresponds to 19 neurons.

**Table 9.** Selection of optimal neurons number (turbo fan, extended model).

| Fault Parameter | Neurons Number in the 2nd Hidden Layer | | | | | | | |
|---|---|---|---|---|---|---|---|---|
| | 14 | 15 | 16 | 17 | 18 | 19 | 20 | 21 |
| ΔLPCC | 1.2903 | 1.2797 | 1.2820 | 1.2674 | 1.2270 | 1.2857 | 1.2719 | 1.2752 |
| ΔLPCE | 0.5195 | 0.5141 | 0.5153 | 0.5161 | 0.5536 | 0.5196 | 0.5286 | 0.5093 |
| ΔHPCC | 0.1842 | 0.1994 | 0.1884 | 0.1976 | 0.2129 | 0.1798 | 0.1928 | 0.1863 |
| ΔHPCE | 0.2311 | 0.2453 | 0.2385 | 0.2359 | 0.2469 | 0.2301 | 0.2263 | 0.2356 |
| ΔHPTC | 0.1431 | 0.1500 | 0.1527 | 0.1623 | 0.1418 | 0.1498 | 0.1571 | 0.1524 |
| ΔHPTE | 0.1787 | 0.1891 | 0.1810 | 0.1818 | 0.1865 | 0.1715 | 0.1798 | 0.1802 |
| ΔLPTC | 0.3961 | 0.3930 | 0.3945 | 0.3858 | 0.4114 | 0.3890 | 0.3920 | 0.3872 |
| ΔLPTE | 0.2981 | 0.2964 | 0.2963 | 0.2953 | 0.2885 | 0.2941 | 0.3149 | 0.2970 |
| Average | 0.4051 | 0.4084 | 0.4061 | 0.4053 | 0.4086 | 0.4024 | 0.4079 | 0.4029 |
| Fault Parameter | Neurons Number in the 2nd Hidden Layer | | | | | | | |
| | 22 | 23 | 24 | 25 | 26 | 27 | 28 | 29 |
| ΔLPCC | 1.2641 | 1.2656 | 1.2728 | 1.2309 | 1.2807 | 1.2801 | 1.2988 | 1.2191 |
| ΔLPCE | 0.5364 | 0.5296 | 0.5524 | 0.5251 | 0.5338 | 0.5454 | 0.5469 | 0.5530 |
| ΔHPCC | 0.1806 | 0.1866 | 0.1862 | 0.2084 | 0.2093 | 0.2033 | 0.1945 | 0.2092 |
| ΔHPCE | 0.2405 | 0.2464 | 0.2405 | 0.2378 | 0.2531 | 0.2370 | 0.2338 | 0.2434 |
| ΔHPTC | 0.1494 | 0.1492 | 0.1499 | 0.1562 | 0.1559 | 0.1447 | 0.1488 | 0.1570 |
| ΔHPTE | 0.1761 | 0.1849 | 0.1841 | 0.1964 | 0.1738 | 0.1886 | 0.1745 | 0.1890 |
| ΔLPTC | 0.3949 | 0.3958 | 0.3933 | 0.3797 | 0.3870 | 0.3933 | 0.3842 | 0.3816 |
| ΔLPTE | 0.2984 | 0.2891 | 0.2946 | 0.2970 | 0.3043 | 0.2898 | 0.2981 | 0.2887 |
| Average | 0.4050 | 0.4059 | 0.4092 | 0.4039 | 0.4122 | 0.4103 | 0.4100 | 0.4051 |

*4.3. Comparison between MLP- and Polynomials-Based Models*

To find the best model and clarify whether the described above accuracy loss is related to the used network or is a common problem, a polynomial-based turbo fan diagnostic model was also developed. Table 10 shows the RMSEs of both models. The polynomials have the same accuracy problem. Moreover, their errors are greater for all fault parameters and almost double on average in comparison with MLP.

**Table 10.** Fault parameter RMSEs of MLP and polynomials (turbo fan).

| Fault Parameter | MLP | Polynomials |
|---|---|---|
| ΔLPCC | 1.2857 | 1.3230 |
| ΔLPCE | 0.5196 | 1.1355 |
| ΔHPCC | 0.1798 | 0.6951 |
| ΔHPCE | 0.2301 | 0.6106 |
| ΔHPTC | 0.1498 | 0.4753 |
| ΔHPTE | 0.1715 | 0.5401 |
| ΔLPTC | 0.3890 | 0.6701 |
| ΔLPTE | 0.2941 | 0.5859 |
| Average | 0.4024 | 0.7544 |

To gain a better understanding of fault parameter behavior and errors, the below analysis uses a graphical mode. Since the first two fault parameters corresponding to a ventilator expose greater errors, Figure 7 presents them separately and Figure 8 illustrates the parameters of the other components, namely, HPC, HPT, and LPT. These figures plot three quantities for each parameter: original values generated by Gas Turb, MLP simulations, and polynomials simulations.

One can see in Figure 7 that there are visible differences (errors) between true and simulated values for both parameters and both models. The first parameter of Delta LPC

Capacity has greater errors for both models, while for the second parameter MLP is more accurate. Additionally, the plots show that the simulation of the first parameter by both models becomes worse after point 100 where an operating regime changes.

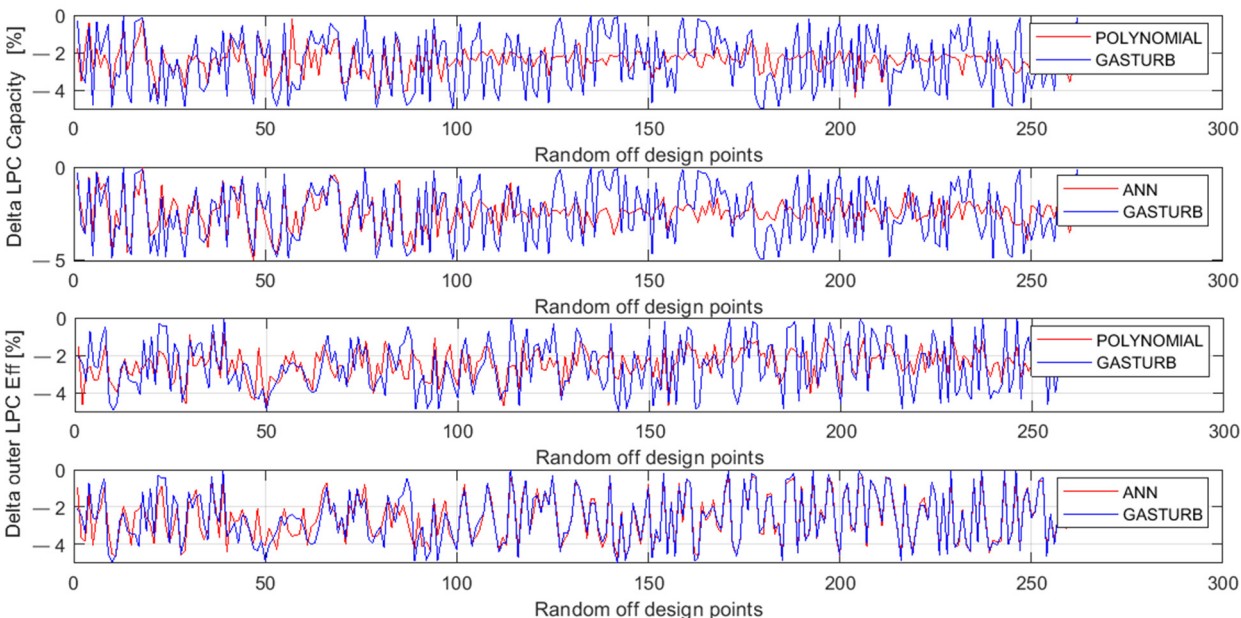

**Figure 7.** Ventilator fault parameters generated by GasTurb and approximated by the polynomials and MLP.

Figure 8 helps to analyze the behavior of the rest of the fault parameters. A general impression of the plots presented here is that the simulated values satisfactorily follow original ones. All errors are significantly lower than those of the ventilator parameters. The only exception is the first parameter "Delta HPC Capacity" inaccurately simulated by polynomials. Moreover, for each of the parameters presented, polynomials yield to MLP.

The comparison of the two engine models shows that the turbo fan model accuracy is mostly acceptable but has worsened for both the MLP- and polynomials-based models in comparison with the turbo shaft models. This worsening is primarily related with great simulation errors of the ventilator fault parameters. Since these errors take place for both approximation techniques, they are not a particular technique problem and rather present a common estimation problem. Probable explanations are the low influence of these parameters on monitored variables and the increase of the total number of estimated parameters (from 6 for the turbo shaft to 8). Additional negative factors are the decrease of the number of monitored variables from 8 to 7 and the considerable reduction of learning data from about 6200 to 3000 operating points.

To solve this error problem, it is natural to exclude LPC fault parameters from the estimated quantities because LPC is a ventilator that is open to a visual inspection. Another way is a multipoint diagnostic option. Within this option, a diagnostic model will get the measurements from different operating points for estimating a single fault parameter vector. Such an increase of input information will yield significant improvement of estimation accuracy.

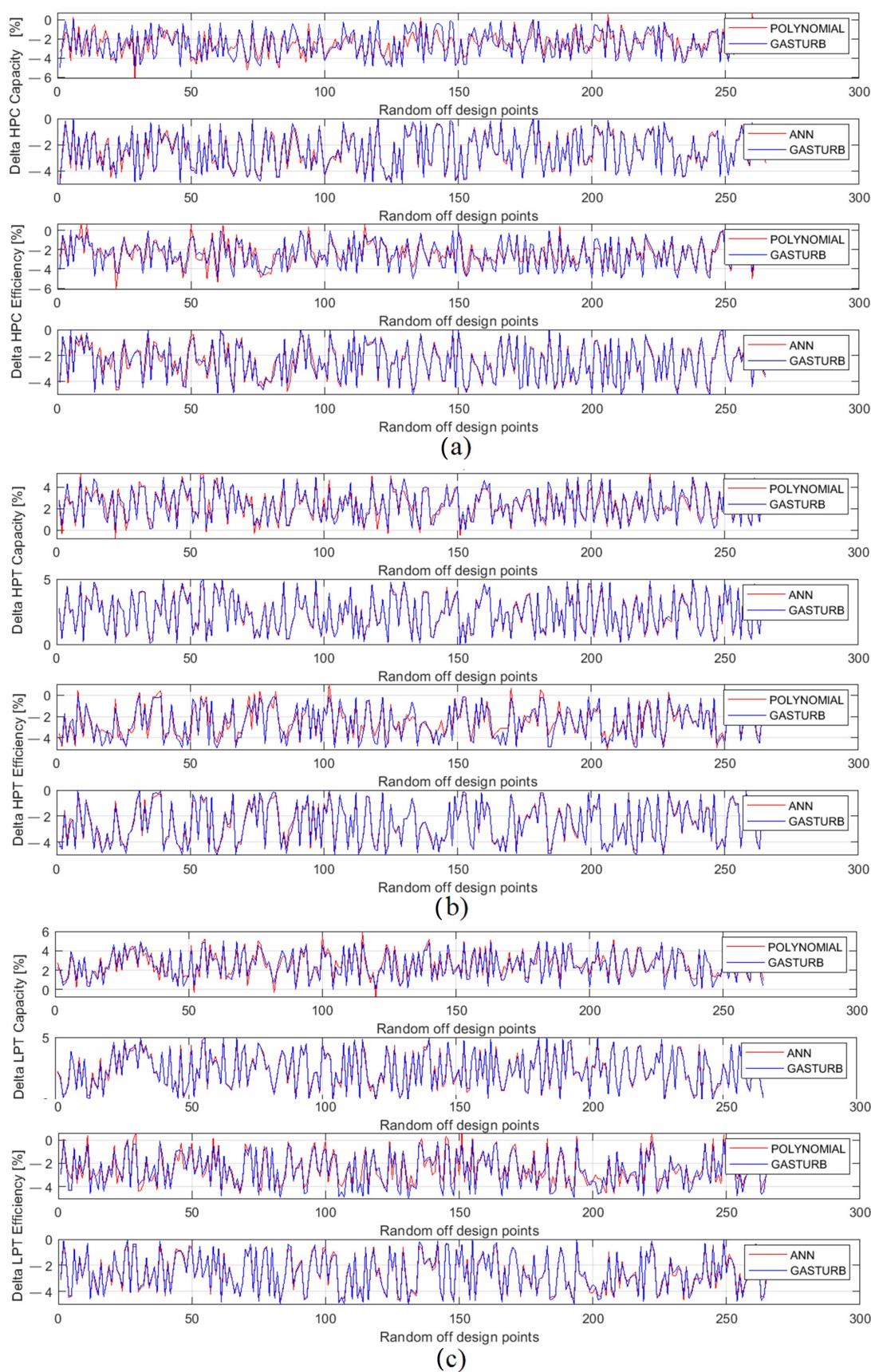

**Figure 8.** HPC, HPT, and LPT fault parameters generated by GasTurb and approximated by polynomials and MLP: (**a**) high pressure compressor parameters, (**b**) high pressure turbine parameters, (**c**) power turbine parameters.

## 5. Conclusions

Inverse gas turbine models were developed and tested in the present paper. These diagnostic models compute unknown fault parameters of each engine component and in this way provide useful information about component health conditions. Since the goal of the GPA diagnostic approach is reached, the developed models present a complete method to diagnose gas turbines.

To obtain more general and reliable results, two alternative approximation techniques, artificial neural networks (namely, MLP) and polynomials, were chosen and applied to create the models of two different engines, turbo shaft and turbo fan. In general for all cases considered, it was found that the accuracy of fault parameters was sufficient for successful engine diagnosis. Although the turbo fan models have greater approximation errors, these errors remain significantly lower than the true values of the fault parameters. This allows correct evaluation of the health of engine components. As to the approximation techniques, a two hidden layers MLP was found to be the best and is recommended for further use.

In contrast to an original nonlinear GPA that employs a physics-based nonlinear model and an iterative adaption procedure for estimating fault parameters, the new data-driven models do the same job, but by far faster and more reliably. Since the developed models present simple, fast, and reliable diagnostic algorithms and have high accuracy, they can receive a wide application in real online diagnostic systems.

**Author Contributions:** Conceptualization, I.G.C. and I.L.; methodology, I.G.C. and I.L.; software, I.G.C.; validation, and J.L.P.R.; formal analysis, I.G.C.; investigation, I.G.C.; resources, I.G.C. and I.L.; data curation, J.L.P.R.; writing—original draft preparation, I.G.C.; writing—review and editing, I.L.; visualization, I.G.C.; supervision, I.L.; project administration, I.L. All authors have read and agreed to the published version of the manuscript.

**Funding:** This research received no external funding.

**Informed Consent Statement:** Not applicable.

**Data Availability Statement:** Not applicable.

**Acknoledgements:** The authors had the support of the Instituto Politécnico Nacional (México) and UNAM-DGAPA through the Postdoctoral fellowship program.

**Conflicts of Interest:** The authors declare no conflict of interest.

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
