# Peer review of "Data-Driven Models for Gas Turbine Online Diagnosis"

_machines, doi:10.3390/machines9120372_

Round 1

Reviewer 1 Report

The authors continue their interesting work on gas turbine diagnostics.

Although the work is interesting, the originality is not completely clear, since non-linear data driven models are used for control. Some relevant papers can be cited and a more elaborated discussion would be welcomed. Some parts of the introduction are text book and can be omitted.

The wording in some cases is not paper appropriate (e.g. "Let us now describe the second approach").

The methodology is consistent, in accordance with the state of the art of the discipline.

It is recommended that the authors support more the originality of their work and review the language after a second reading. 

Author Response

The response is in the file attached

Reviewer 2 Report

This manuscript (Title: Data-Driven Models for Gas Turbine Online Diagnosis; Manuscript ID: machines-1455486  ï¼‰studies the data-driven model of gas turbine.  Polynomials and neural networks are adopted. According to the content presented, the following aspects need to be improved.
1. The resolution of picture 1 needs to be enhanced.
2. The parameters and data in Table 2 are somewhat disordered.
3. Line 237, the expression of data range (the interval [0% - 5%],) is not standardized.  Check other range dimensions.
4. In Section 2, the description of the method lacks mathematical support.
5. In Figure 2, the text box used to distinguish lines should not obscure the data curve. More local detail windows can be added to the two comparison curves to observe the gap.
6. The references are somewhat old. Please refer to and quote the latest research.

Author Response

The response is in the file attached

Round 2

Reviewer 2 Report

1. For this manuscript (ID: machines-1455486 ), the author has completed the corresponding modification. The research content is substantial and detailed.
2. In Table 2, the display of the first column in PDF format is still garbled, please adjust it.